# Effect of Blue Light on Acne Vulgaris: A Systematic Review

**DOI:** 10.3390/s21206943

**Published:** 2021-10-19

**Authors:** Mara Lúcia Gonçalves Diogo, Thalita Molinos Campos, Elsa Susana Reis Fonseca, Christiane Pavani, Anna Carolina Ratto Tempestini Horliana, Kristianne Porta Santos Fernandes, Sandra Kalil Bussadori, Francisca Goreth Malheiro Moraes Fantin, Diego Portes Vieira Leite, Ângela Toshie Araki Yamamoto, Ricardo Scarparo Navarro, Lara Jansiski Motta

**Affiliations:** 1Biophotonics Applied to Health Sciences Department, Nove de Julho University (UNINOVE), Vergueiro 245, São Paulo 01504-001, Brazil; dermatomara3@gmail.com (M.L.G.D.); thalitamolinos@gmail.com (T.M.C.); chrispavani@gmail.com (C.P.); acrth@gmail.com (A.C.R.T.H.); kristianneporta@gmail.com (K.P.S.F.); sandrakalil.skb@gmail.com (S.K.B.); 2Physics Department Covilhã, Universidade da Beira Interior, 6200-354 Covilhã, Portugal; ersfonseca@gmail.com; 3Department of Bioengineering, Universidade Brasil, São Paulo 08230-030, Brazil; g.p.fantini@terra.com.br (F.G.M.M.F.); odontoportescursos@gmail.com (D.P.V.L.); ricardosnavarro@gmail.com (R.S.N.); 4Dentistry Department, Universidade Cruzeiro do Sul, São Paulo 01311-925, Brazil; a_araki@me.com

**Keywords:** acne, photobiomodulation, LLLT (Low Level Light Therapy), phototherapy, LED (Light Emitting Diode), blue light

## Abstract

Acne is a dermatosis that affects almost 90% of the adolescent population worldwide and its treatment is performed with retinoids, antimicrobials, acids, and topical or systemic antibiotics. Side effects such as skin irritation in addition to microbial resistance to antibiotics are the main side effects found. Phototherapy with blue light is being used as an alternative treatment. Our objective was to analyze the use of blue light to treat inflammatory acne. We conducted a systematic literature review, following the recommendation PRISMA (Preferred Reporting Items for Systematic Reviews and MetaAnalyses), including in the sample randomized clinical trial studies that compared blue light with another intervention as control. The research was carried out in the PUBMED and WEB of SCIENCE databases and the methodological quality of the studies evaluated were made by the Cochrane Collaboration Bias Risk Scale. After the exclusion of duplicates, the titles and abstracts of 81 articles were evaluated, and 50 articles were selected for full reading, including in the review at the end 8 articles. Studies have shown significant improvements in the overall picture of acne. It is concluded that despite the great potential in its use in the treatment of acne, there is a need for more detailed trials on the effect of blue light on the treatment of inflammatory acne.

## 1. Introduction

Inflammatory acne is the main cause of the search for dermatological medical offices in Brazil [1] and worldwide [2,3]. It is estimated that approximately 10% of the world population (650 million people) is affected by the disease, being the eighth type of disease, in prevalence, in the world [2,3]. Its consequences are scars, post-inflammatory hyperpigmentation, as well as psychological damage such as depression and anguish, leading to the individual’s removal from social life, depending on the degree of severity [4,5,6,7].

Acne lesions can be considered inflammatory and non-inflammatory. Non-inflammatory lesions are known as microcomedones, not visible to the naked eye. Microcomedones can form due to factors such as linoleic acid deficiency, excessive secretion of androgens, or excess free fatty acids [8]. Comedones can appear as closed comedones—white papules smaller than 0.5 cm in diameter—and open comedones—black spots, pigmented by melanin, presenting keratin and lipid deposit in the follicle, which is considered the standard lesion of acne [9].

Inflammatory lesions can be (a) Papule—arising as an area of erythema and edema up to 5 mm; (b) pustule—punctiform inflammatory lesions with yellowish secretion in the center; (c) nodule—inflammatory lesions larger than 5 mm in diameter; (d) cyst—great comedones that undergo several ruptures and encapsulations, presenting tense, protruding globosus, with pasty and caseous content, and (e) scar—irregular depression covered with atrophic skin, finely telangiectatic, resulting from the destruction of the hair-sebaceous follicle by inflammatory reaction [8,9].

There are several scales of classification of the degree of acne worldwide which are used in the diagnosis of the disease [9,10]. Here are a few:

The Simple Grading of Acne [11], published by the Global Alliance to Improve Outcomes in Acne (2003), classifies lesions as Grade I, or non-inflammatory, in which they have only closed and open comedones; Grade II, which, in addition to many comedones, there are also papules and pustules, causing inflammatory lesions; Grade III, which presents a mixture of papules, pustules, comedones, nodules and cysts; Grade IV, which in addition to all previous factors, also presents fistulas and abscesses, affecting its severity with lesions that evolve to scar processes.

The IGA (Investigator Global Assessment) scale, recommended by the FDA (Federal Drug Administration), 2005 [11], follows the diagnostic criteria: Grade 0—Clean skin, without inflammatory and non-inflammatory lesions; Grade 1—Almost clean skin, with few non-inflammatory lesions and no more than one inflammatory lesion; Grade 2—Mild severity, higher than grade 1, with some non-inflammatory lesions and few inflammatory lesions (papules and pustules without nodular lesions); Grade 3—Moderate severity, many non-inflammatory lesions, some inflammatory lesions and no more than a small nodular lesion; Grade 4—Severe, higher than grade 3, many non-inflammatory and inflammatory lesions, some nodular lesions. The FDA does not recommend the use of Grade 5, as other authors indicate in the literature [12].

The classification of acne in Brazil is based on the description of the SBD (Brazilian Society of Dermatology), with the following parameters [13]:

Grade I, comedones, without inflammatory lesions (Figure 1A); Grade II, comedones, papules, and pustules with varying intensity and few to numerous inflammatory lesions with some erythema (Figure 1B); Grade III, comedones, papules, and pustules with an intense inflammatory reaction that leads to the formation of nodules, which may contain pus (cysts) (Figure 1C); Grade IV, comedones, papules, pustules and fistulas, larger cysts forming large lesions [14].

Acne can be treated by intervention techniques that aim to eliminate inflammation and non-inflammatory visible lesions in the acute phase of the disease. Maintenance procedures to minimize relapse and adjuvant treatments to improve skin appearances such as scarring and post-inflammatory hyperpigmentation. Ideally, treatments need to have minimal side effects and acceptable tolerability [15].

For the treatment of Grade I and Grade II acne, topical products are generally used. The most common are retinoids, such as adapalene, retinoic acid, isotretinoin, which have anti-comedogenic, anti-inflammatory, and comedolytic characteristics. The main disadvantage of most topical retinoids is related to cutaneous side effects observed in up to 75% of patients, including erythema, flaking, dryness, burning, and itching. In addition, certain formulations of topical retinoids are inactivated by sunlight in addition to being contraindicated in pregnancy, and women of childbearing age should use effective contraceptive methods during treatment because they have teratogenic characteristics [16,17].

In the cases of Grade III and IV acne, in addition to topical products, oral contraceptives, antiandrogenics, and oral antibiotics such as doxycycline, minocycline, tetracycline [17,18] are used. In the case of antibiotics, besides the hepatotoxic effects, there is the problem of microbial resistance. For approximately 94% of acne cases, P. acne (Propionibacterium acne) or currently C. acne (Cutibacterium acne), observed on the skin are resistant to at least one antibiotic [16]. More than 50% of C. acnes are resistant to erythromycin in Egypt, France, Greece, Italy, Spain, and the United Kingdom and clindamycin in Egypt, Greece, Hong Kong, Italy, and Spain, which reduces the effectiveness of antibiotic acne treatment but can also spread to untreated contacts and therefore affect more antibiotic resistance patterns in the population. And for oral retinoids, there is contraindication in the reproductive phase because they are teratogenic, in addition to hepatotoxic [19]. 

Considering the need for treatments with fewer side effects, alternatives have been presented over the years. An example of a safe alternative is phototherapy.

According to the literature, phototherapy and photobiomodulation are synonymous, but there is a preference for photobiomodulation terminology, and the reason is understandable because biomodulation can mean biostimulating or bio inhibiting, depending on the optical properties of tissue and the dosimetric parameters of optical radiation [20,21]. These parameters should include at least the type of source used (Laser, LED, light, etc.), the wavelength (Nanometers), the power (W), the energy (J), the radiant exposure (J/cm^2^), the creep (J/cm^2^) the irradiance (W/cm^2^)and the mode of application of the device (contact, punctual, distance).

The most used light sources for photobiomodulation are lasers (Light Amplification by Stimulated Emission of Radiation) and LEDs (Light Emitting Diodes).

The bacterium C. acnes produce porphyrins [22] that absorb light energy in the spectrum of ultraviolet and blue light. The evaluation of the effect of blue light on acne treatment demonstrated that irradiation of colonies of C. acnes with visible blue light LED to photoexcitation of bacterial porphyrins and production of singlet oxygen and eventually bacterial destruction, indicating that acne can be successfully treated with phototherapy with blue visible light [23].

Some studies have been conducted with blue light in the treatment of acne as an example:

A multicenter, randomized study treated 89 people in hemiface, half of them with led blue light phototherapy (446 nm) and photoconverter chromophores twice a week for 6 weeks and was shown to be effective when compared to untreated hemifaces, with a significant reduction of at least 40% in inflammatory acne lesions. These participants were followed for another 6 weeks and at week 12 there was an even greater difference in the hemiface treated with blue light (81.6% treated vs. 46.0% in control (*p* < 0.0001) [24].

Another study treated thirty people with blue light (407 to 420 nm), (8 times, twice a week) and the other thirty were treated with a topical formulation of Benzoyl Peroxide 5%, self-applied twice a day, every day. The improvement achieved with blue light was equal to that of benzoyl peroxide, regardless of the type of lesion (*p* = 0.05). However, side effects were less frequent in the group treated with blue light [25].

Domestic therapy with blue light was evaluated using a device with a wavelength of 414 nm and after 3 weeks there was a significant decrease in the inflammatory lesion and erythema [25]. The skin presented a smoother texture and increased tone [26]. Some authors state that inflammatory lesions respond better to blue light phototherapy than non-inflammatory lesions [27].

The market for light therapy devices for acne is growing and the patient’s interest in these devices is increasing [28]. However, there is no consensus about the ideal parameters for the treatment of acne with light [29].

Considering that acne negatively affects the quality of life, in the self-perception, social and emotional dimensions, and still presents itself as a universal disease, we believe that it should be treated by a multidisciplinary health team [30], which should present alternatives, through scientific research, both for treatment and to minimize the side effects of the disease. The objective of this research was to review the researches that used blue light in the treatment of acne, evaluating their quality.

## 2. Materials and Methods

The methodological procedures followed the guidelines proposed by the international network Enhancing the Quality and Transparency of Health Research (EQUATOR). The project was registered in the OSF (Open Science Framework) platform, Registration 10.17605/OSMIO/U62 XF. Motta, 2020 Lara J. “TREATMENT OF ACNE VULGAR WITH LASER/BLUE LED: A SYSTEMATIC REVIEW”.

The research was developed as a product of the research line of the project coordinator, permanent professor of the Graduate Program in Biophotonics Applied to Health Sciences of the Nove de Julho University in São Paulo, SP.

The bibliographic search followed the development of the SR protocol (Systematic Review) of the literature and will follow the Preferred Reporting items for Systematic Reviews and MetaAnalyses (PRISMA) guidelines (Figure 2) which provides a checklist for transparency in the selection process of articles.

The research question (Patient, Intervention, Comparison and Outcomes/Outcome (PICO), in the Table 1, was as follows: What is the clinical evidence for the action of blue light in improving acne concerning conventional treatments?

The searches were carried out at PubMed (8–10 April 2021), Cochrane (13–15 April 2021), and Scopus (19–21 April 2021) databases, and the following terms and combinations were used: (Acne) and (photobiomodulation) or (LLLT) Low Level Light Therapy or (Phototherapy) or (LED) Light Emitting Diode and (blue light). After the search, the titles and abstracts of the articles were independently selected by two reviewers, and in the case of disagreement, a third evaluator determined the inclusion or not.

The following inclusion criteria were applied:Only randomized controlled trials from 1990 to 2021;Articles that presented control group.

And the following exclusion criteria:Duplicates or studies with the same number of ethical approval.

## 3. Results

In the electronic search, we found 6745 articles with the same keywords as the ones in this study. When we applied the filters seeking randomized controlled studies, we found 2002 articles and thus removing the duplicates and selecting articles that included blue light and a control group, 8 articles remained (Table 2).

Antoniou et al. [24], conducted a study with 89 patients divided into 2 groups, using treated Hemiface, chosen by computer-generated listing, applying a photo converter gel chromophore and then a multi-LED device with a wavelength from 415 to 446 nm, applied for 5 min at a distance of 5 cm, and the other half of the face, which was not treated, was used as a control group. The dose used was 33 to 35 J/cm^2^ and irradiance was 110 and 150 W/cm^2^. Two weekly treatment sessions were performed for 6 weeks and for another 6 weeks the patients were followed up after treatment. The degrees of acne using the IGA scale at the beginning and end of treatment and the count of lesions were evaluated.

According to the results of this study, there was a reduction of at least two degrees in the severity of acne according to the IGA scale, which was demonstrated in 51.7% of patients at week 12, in the light-treated group. In addition, at week 12, individuals with a grade 3 (moderate) baseline IGA had a drop of 2 degrees or more in the degree of acne (45.3%), while patients with a baseline IGA degree of 4 (severe) demonstrated a success rate of 61.1%. The number of inflammatory acne lesions dropped at least 40% in 81.6% of the hemifaces treated after 12 weeks. In the control group, with an untreated face, only 18% reached a reduction of 2 degrees in the acne scale. The comparison of Cardiff Disability Index (CADI) scores, a patient satisfaction questionnaire concerning the improvement in acne, which was applied by the authors, indicated a 40% decrease in hemifaces treated at weeks 6 and 12, while an increase in CADI scores of 20% was observed for the untreated group.

In a randomized open-air clinical study, Arruda et al. [25], compared the efficacy of blue light with benzoyl peroxide (BPO) at 5% in the treatment of inflammatory acne of grades 2 and 3. The study evaluated 60 patients divided into 2 groups of 30 patients, finishing with 28 patients in the BPO group and 24 in the blue light group, through lesion count and photographs. The wavelength used for blue light was 407 to 420 nm, in eight sessions, applied twice a week. Irradiance was 40 mW/cm^2^ and the opening diameter of the appliance was 55 mm. The BPO group used the cream twice a day, daily for 28 days. The results showed that there was a reduction of 31.32% in inflammatory and non-inflammatory lesions as BPO based treatment and in the blue light group there was a reduction of 21.66% in lesions (inflammatory and non-inflammatory) after treatment. However, the report of adverse effects in the BPO group was 93.3% and there was a need to reduce the number of daily applications of the product, while in the blue light group, there was a complaint by 23.3% of patients with adverse effects.

Cheema et al. [31], compared 124 patients with mild to moderate acne in a controlled and randomized study, divided into 2 groups, one treated with blue light, wavelength between 407 and 420 nm, and the size of the spot opening of the device of 55 mm, and the other with BPO at 4%. The evaluation was made by counting lesions and the evaluation of the severity of acne determined by the number of lesions: Acne was classified as mild acne: less than 20 comedones, less than 15 inflammatory lesions, or total count of lesions less than 30; Moderate acne: between 20 and 100 comedones, 15 to 50 inflammatory lesions or total count of lesions from 30 to 125. Patients in the blue group received treatment for 15 min in each session, twice a week, for 6 weeks, and patients in the BPO group used the product every night for 6 weeks. There was a greater reduction in lesions in the blue light group compared to the BPO treated group (76% × 60%). Another finding in this study was the reduced adverse effects in the blue light group when compared to BPO.

Elman et al. [32], analyzed blue light, in the range of 405 to 420 nm, in papulopustular acne in a study divided into 3 parts: the study of the split face (half face), 10 patients were treated where the left side was exposed to light for 8 min and the right side for 12 min. The form of application of the device was in contact with the skin. There was a decrease in the mean number of 21 inflammatory lesions per patient before therapy to 7.7 after eight treatment sessions. The decrease on the left side averaged 65.9%, and on the right side of the face averaged 67.6%.

In the complete face study, with open evaluation, 13 patients were treated with light for 15 min, twice a week, for 4 weeks. There was a reduction of 81% of inflammatory lesions. In the split-face, double-blind, self-controlled study, one side of the face was chosen to be treated for 15 min, twice a week, for 4 weeks. 23 patients were treated and evaluated by a blind doctor. There was a 20% reduction in inflammatory lesions and an average of 60% reduction of lesions. The untreated side had an average reduction of 30% in inflammatory lesions.

Gold et al. [26], evaluated 30 patients using on one side of the face a domestic device with blue light (414 nm), and the other side a sham device. The lesions to be treated were similar, with one on each side of the face, and randomly chosen. The lesions were evaluated using the following criteria: lesion size (not elevated, mild, moderate, or severely elevated) and erythema (none, trait, moderate, severe). After 4 applications there was an improvement in the size of the lesion of 76% with the device compared to only 41% with the sham device. There was an improvement in the treated group of 37% compared to 10% in the control group.

Kwon et al. [33], conducted a study with 24 patients evaluating inflammatory acne, through injury count and sebum measurement, comparing blue light (450 nm) associated with 1450 nm laser diode and blue light alone on hemifaces during 3 sessions with an interval of 4 weeks. The blue light was applied for 3 min with an irradiance of 3.5 to 7 mW/cm^2^ and a dose of 0.6 to 1.2 J/cm^2^. There was an improvement of 62.3% and 35.2% for the combination of blue light and diode laser and blue light, respectively, in the inflammatory lesions of acne. However, the improvement of seborrhea was verified with the combined laser diode lights (1450 nm) associated with blue light.

In the study by Papageorgiou et al. [34], a comparison was made of the blue light (415 nm), blue and red combined, white light, and benzoyl peroxide in cream, with 107 patients divided into these 4 groups. The device was applied for 15 min daily for 12 weeks at a distance of 25 cm, with a dose of 320 J/cm^2^ and 4.23 mW/cm^2^ of irradiance. Although the results favored the combined group of blue and red light with a small advantage, there was an improvement in inflammatory lesions with the blue light of 63% and in the commedones of 45%.

And Tzung et al. [35], conducted a study with 31 patients on one side of the face with blue light, leaving the other face, without any treatment, as a control. A score was given for each type of acneic lesions such as commedones, papules, pustules, nodules, and cysts and the scars were also scored before treatment. The evaluations were made with Wood’s lamp before and after treatment. After 8 sessions, done twice a week with a wavelength of 420 +/− 20 nm, applied at 15 cm distance, with a dose of 40 J/cm^2^ per session, 320 J/cm^2^ in total, there was an improvement of 52% in the acne condition compared to 15% the untreated face. However, it was found that among the characteristics of acne, acne populous—pustules improve more with the blue light treatment concerning comedonian and nodulocystic acne, and also, the size of the pore can not be used as a predictive factor of therapeutic efficacy.

The dosimetry parameters of the studies are described in Table 3.

### Risk of Bias

The articles with the lowest risk of bias were Papageorgiou et al. [34]. and Kwon et al. [33] and the one with the highest risk of bias was Arruda et al. [25] because it included patients in order of care for the treatments performed and does not mention the blinding of professionals and evaluators, besides having had a significant loss of participants in the groups, which may have influenced the results of the research. However, in most studies there was no information on how the selection of participants was made or even the selection was made to suggest an uncertain risk of bias. Most studies also did not or did not report the way the participants were allocated in the research groups, as well as the blinding of the participants or professionals who applied the treatments and those who evaluated the results (Table 4).

## 4. Discussion

To answer the question “What is the clinical evidence for the action of blue light in improving acne condition when compared to conventional treatments?” from the PICO research conducted in the systematic review, we found a number of scientific articles that proved the efficacy of treatments with LED and blue light devices in the effective improvement of acne and with minor or non-existent adverse effects compared to conventional treatments such as BPO.

In the articles with lower risk of bias, there was an improvement of 35.30% in acne lesions (Kwon et al. [33]) and in Papageorgiou et al. [34], there was an average improvement of 54% in acne lesions, including comedones and papules with the treatment with blue light. Although the comparative side treated with blue light, in the study by Kwon et al. [33], had an improvement of 62.30% in acne lesions, in this case, an association of a laser (1450 nm) was used, which should be used with caution for risk of causing erythema, moderate to severe pain and post-inflammatory hyperpigmentation [36].

In the study by Papargeorgiou et al. [34], the red light associated with blue light had a better result in the improvement of acne lesions at first (76%), which did not occur at the end of treatment. Even so, blue light alone was also quite effective at the end of treatment, with an average improvement of 54% in acne lesions (45% in comedones and 63% in inflammatory lesions).

This result corroborates most studies that assess the effect of blue light on acne treatment and demonstrate that irradiation of Cutibacterium acnes colonies with visible blue light led to photoexcitation of bacterial porphyrins, singlet oxygen production and eventually bacterial destruction, indicating that acne can be successfully treated with blue visible light phototherapy.

It is known that the associated treatments have greater effectiveness in the outcome of acne treatment and that C. acnes, a gram-positive bacterium, can develop resistance to topical and systemic treatments commonly used in anti-acne treatment, generating global health impact [37]. Topical retinoids or benzoyl peroxide should be administered daily, which may cause skin irritation and lead to low patient support and thus promoting ineffective results. The use of isotretinoin requires rigorous professional monitoring and may result in adverse events [38]. Therefore it is essential to explore new therapeutically effective treatment models for acne.

Light therapies have emerged as an alternative that offers a unique type of treatment for acne. Currently, treatments for mild to moderate inflammatory acne include a variety of high irradiance light technologies, such as intense pulsed light (IPL) and photodynamic therapy (PDT) [39,40]. Photobiomodulation (PBM) is a non-thermal light therapy used to treat acne and other dermatological conditions [41]. Irradiation levels employed by LEDs or lasers in PBM treatments are considerably lower than in ablative treatments and work by activating biologically active photo pathways in their target tissues. To have any effect on a living organism, therapeutic photons must first be absorbed by a molecular or photoreceptor chromophore, such as porphyrins, flavins, or other light absorbers within the cell [20,21]. Therefore, dosimetric parameters must be studied and established to facilitate the work of health professionals who use these therapies as a treatment for acne.

It is important to note that all the selected studies had different parameters in studies and, in many of them, the parameters were not described correctly, or were incomplete, which prevents us from reproducing these studies. We verified that the wavelength used ranged from 405 to 450 nm, which, theoretically, would be in violet color, a discussion worth attention because all the studies that used this wavelength describe them as blue light. The form of light application was mentioned only in 3 studies. It is known that the way light is applied into tissues directly impacts its penetration, which is governed by both absorption and scattering by molecules and structures present in the tissue [42]. This is fundamental because in order to react against the tissues light needs to be absorbed by the target molecule. The first law of photochemistry states that “light must be absorbed before photochemistry occurs” [43].

Other important parameters have not been described by some authors such as the time of application of light, the average power of the device used and the mode of operation of the device, the treated area. The applied energy was described by some authors, as well as irradiance, which are fundamental characteristics to correctly reproduce the studies [44].

Although skin improvement assessments in the studies described were analyzed by dermatologists, another important point was the outcomes used by the authors. Some authors relied on the count of acne lesions and degree of acne, others on the aspect of the lesions, such as erythema, edema, and size of inflammatory lesions, which made it difficult to review the articles to reach meta-analysis. As the evaluation of acne is made through scales, and as there are several scales worldwide, there is no consensus for the assessment of the severity of the condition, which ends up being a clinical and subjective evaluation, with each author describing the characteristics of the disease according to his country of origin and also according to his clinical experience.

Some systematic reviews have been made in recent years on this topic, such as Scott [29], but we think it was important to make a specific review of blue light in the treatment of acne since this review included the red and infrared lights in her study, which was not our goal.

## 5. Conclusions

In conclusion, we found that blue light demonstrated improvement in the treatment of acne, especially in inflammatory lesions and seborrhea, with no significant improvement in grade 1 acne or with only comedones, and not showing scars, null or minimal side effects and that it is a safe alternative to conventional treatments.

We reinforce the need for new controlled, randomized studies so that appropriate parameters are established to help healthcare professionals who offer this type of treatment to their patients and serve as a parameter for manufacturers of home-use appliances.

## Figures and Tables

**Figure 1 sensors-21-06943-f001:**
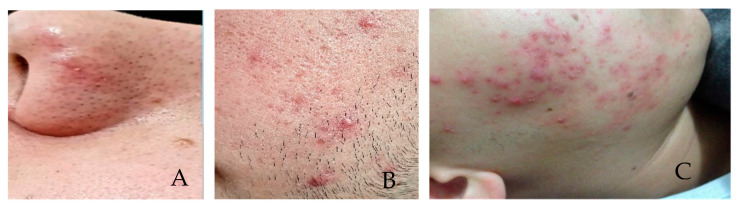
(**A**) Acne—Grade I (**B**) Acne—Grade II—(**C**) Acne Grade III (source: personal file).

**Figure 2 sensors-21-06943-f002:**
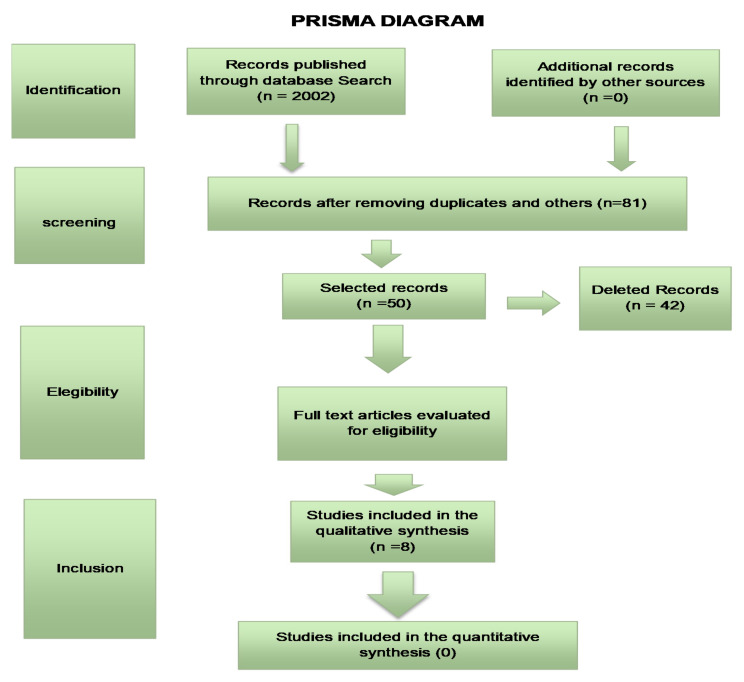
PRISMA diagram according to Bibliographic Search.

**Table 1 sensors-21-06943-t001:** PICO—Patient, Intervention, Comparison and Outcomes/Outcome.

Patient	With Inflammatory Acne
Intervention	Photobiomodulation with blue light
Comparison	Medicines/other therapies
Outcomes	Reduction of lesions

**Table 2 sensors-21-06943-t002:** Characteristics of the studies included in the systematic revie.

Author, Year, Country	Randomization Unit	Groups	Follow-Ups (Weeks)	Average Age (in Years)	Acne Severity Degree	N Total	Light/Wavelength (Blue)	Total No. of Sessions	Duration of Exposure (Weeks)
Antoniou et al., 2016 (Greece)	Half-face	2 groups: 44 Half-face L 45 Half-face D	12	21 (16–30)	Moderate, severe	98	415/446 nm	12	(6) Two weekly treatment sessions were performed for 6 weeks.
Arruda et al., 2009 (Brazil)	Individual	2 groups: 24 blue light and 28 BPO	6	17 (NR)	Moderate, severe	60	407 a 420 nm	8	(4) eight sessions, applied twice a week.
Cheema et al., 2018 (Pakistan)	Individual	2 groups: 62 blue light and 62 BPO	12	23.02 ± 6.3	Mild to moderate	124	407 a 420 nm	12	twice a week, for 6 weeks
Elman et al., 2002 (Israel)	Face/Half-face	3 groups: (1) 10 (Half-face) e (2) 13 (Full-face) (3) 23 (face Blinding)	8	1.8	inflammatory acne lesions	23	405 a 420 nm	8	Goup 1—Not cited Group 2 twice a week for 4 weeks Group 3—twice a week for 4 weeks
Gold et al., 2011 (United States)	Lesion	2 groups: Light and random simulator	After 2 treatments)	30 (NR)	Mild, moderate	30	414 nm	4	4 treatments in 2 consecutive days (2 treatments a day with an interval between 2 and 12 h) for as long as 10 days
Kwon et al., 2019 (korea)	Individual	2 groups: DL + BL/BL (nonspec ific)	12	21.6 ± 7.8	Mild, moderate	24	450 nm	3	3 sessions with an interval of 4 weeks
Papageorgiou et al., 2000 (United Kingdom)	Individual	4: 27 (B) 30 (B + R) 25 (Whith light) 25 (BPO)	12	23, 25, 27 d (NR)	Mild, moderate	82	415 nm +20/−15 nm	84	15 min daily for 12 weeks
Tzung et al., 2004 (Taiwan)	Half-face	2 groups (treated face and control side)	8	21 (15–32)	Moderate, severe	31	420 ± 20 nm	8	8 sessions, twice a week

**Table 3 sensors-21-06943-t003:** Detailing the protocols presented in clinical trials.

Study	Types Light	Wavalengh	Application	Diameter Laser	Application Time	Radiant Exposure	Irradiance	Outcome Variable	Percentage of Treated Side Improvement	Control Side Improvement Percentage
Antoniou, 2016	LED	415/446 nm	5 cm	-	5 min	33 a 35 J/cm^2^	-	Lesions Count and Acne Severity	40%	18%
Arruda, 2009	Light	407 a 420 nm	-	55 mm lighted área	-	-	-	Lesions count and %	21.66%	31.32%
Cheema, 2018	Light	407 a 420 nm	-	55 mm circular area	15 min	-	-	Lesions Count and Acne Severity	76%	60%
Elman, 2002	Light	405 a 420 nm	contact	-	15 min	-	50 a 200 mW/cm^2^	Lesion count	G1 65.9% E, 67.6% D G2 80%; G3 60%	- - G3 30%
Gold, 2011	Light	414 nm	-	-	-	-	-	Lesions size and Erythema	Lesions size 76% Erythema 37%	Lesion size 41% Erythema 10%
Kwon, 2019	Ligth	450 nm	-	Spot size 6 mm	3 min	0.6 a 1.2 J/cm^2^	3.5 a 7 mW/cm^2^	Inflammatory lesion count	35.30%	62.30%
Papageorgiou, 2000	Lamp	415 nm + −20/−15 nm	25 cm	-	15 min	320 J/cm^2^	4.23 mW/cm^2^	Inflamatory lesions	IL 63% comedones 45%	Red light: 76% IL 58% comedones
Tzung, 2004	Light	420 ± 20 nm	15 cm	-	-	40 J/cm^2^ per session	-	Number and size of lesions	52% improvement in acne	15%

Legenda: LED = Light Emission Diode, nm = Nanometros, L = left side, R = Right side, IL = Infamatory lesions, G1 = Group 1, G2 Group 2, G3 Group 3.

**Table 4 sensors-21-06943-t004:** Risk of individual biases of the eight studies selected for systematic review for each domain of risk assessment of bias in randomized clinical trials by the Cochrane collaboration tool.

	Random Sequence Generation	Allocation Concealment	Blinding of Participants and Professionals	Blinding of Outcome Assessors	Incomplete Outcomes (Losses)	Selective Reporting of Outcome	Other Biases
Antoniou, 2016	LOW	UNCERTAIN	UNCERTAIN	UNCERTAIN	LOW	UNCERTAIN	UNCERTAIN
Arruda, 2009	HIGH	UNCERTAIN	HIGH	HIGH	HIGH	UNCERTAIN	UNCERTAIN
Cheema, 2018	UNCERTAIN	UNCERTAIN	UNCERTAIN	LOW	UNCERTAIN	UNCERTAIN	UNCERTAIN
Elman, 2002	UNCERTAIN	UNCERTAIN	LOW	UNCERTAIN	UNCERTAIN	UNCERTAIN	UNCERTAIN
Gold, 2011	UNCERTAIN	UNCERTAIN	UNCERTAIN	UNCERTAIN	UNCERTAIN	UNCERTAIN	UNCERTAIN
Kwon, 2019	LOW	LOW	UNCERTAIN	LOW	LOW	UNCERTAIN	UNCERTAIN
Papagerorgiou, 2000	LOW	UNCERTAIN	LOW	LOW	LOW	UNCERTAIN	UNCERTAIN
Tzung, 2004	UNCERTAIN	UNCERTAIN	UNCERTAIN	UNCERTAIN	UNCERTAIN	UNCERTAIN	UNCERTAIN

Higgins, J. P. T., Savovic, J., Page, M. J., Sterne, J. A. C., 2016. Revised Cochrane risk of bias tool for randomized trials (RoB 2.0). Creative Commons Attribution-NonCommercialNoDerivatives 4.0 International License.

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
