# Peer review of "Effect of Blue Light on Acne Vulgaris: A Systematic Review"

_sensors, 2021, doi:10.3390/s21206943_

Round 1

Reviewer 1 Report

Thank you for submitting your work, evaluating the effects of blue light in management of  acne vulgaris. It is an interesting topic. I have few suggestions and observations require the authors' consideration: 

  1. In table of dosimetry, there is no records of the spot size diameter of the laser and the number of the cluster in the LED device, whether one wavelength or two wavelength. Could you please address this.
  2. It would be useful for the reader to identify which type of ance vulgaris was better responder to blue light.
  3. Have the studies reported any residual scars after treatment? Please address this
  4. No follow-up reported in the tables for the selected studies. Please address this.
  5. Mentioned the number of the sessions but it is not clear what is the time interval between sessions. Please address this

Author Response

Dear Reviewer,

Thank you for your suggestions, and iI made the corretions in the manuscrit.

Revisor 2:

Firstly, we would like to thank the reviewer for all the considerations regarding our article.

Question 1 : In the keywords, write well Led Emitting Diode

Answer:

Thank your for pointing it out. It has been corrected and highlighted in the text.

Question 2 : The Figure 1 could be more orderly. Especially add the letters in the pictures, as reported in the caption, to facilitate the reader.

The source is referred to as "personal file": has the patient’s consent been requested for the publication of the image? I suggest eliminating the details of the photo that could allow the recognition of the person, like the eyes

Answer:

Thanks again for pointing this out. The pictures have been corrected. I have the patient’s written consent.

Question 3 : Introduction: what is meant by "condition of the target?" perhaps the optical properties of a specific biological tissue? If so, I suggest speaking of "optical properties". Otherwise indicate some of the conditions.

Answer:

Many thanks for the suggestion, which I accepted, as highlighted in the text.

Question 4 : Recheck the text. In particular, always write in the same way "blue light" (sometimes it is uppercase) and "comedones".

Answer:

Many thanks again. I have corrected these slips and I apologize for overlooking them.

Question 5 :  In the clinical studies you have reported, is it possible to write who or how skin improvements have been evaluated following the blue light treatments? in my opinion is an important information that adds value to the result obtained in the study

Answer:

The evaluations were made by dermatologists, in all studies. As acne evaluation is most often clinical, it might be rather subjective, even though there are scales for grading acne occurrences, which might aid an accurate diagnostic, as mentioned in the text.

Question 6 : Correct the title.

Answer:

Thanks again. We have changed the title.

Question 7 : I suggest that authors report the bibliographic reference every time an author is quoted.

Answer:

Thanks. I have revised this. Please accept my apologies.

Thank you for your consideration.

Sincerely,

Mara Lúcia Gonçalves Diogo

Reviewer 2 Report

In this systematic review, Mara Diogo et al. performed research about the treatment of acne vulgaris with blue light. In the emerging context of photobiomodulation, where literature is often confused and it is difficult to compare different articles, it is useful to have systematic reviews as a reference point.

I find the work well structured and I will only make a small list of suggestions/changes that authors may consider making:

  1. In the keywords, write well Led Emitting Diode
  2. The Figure 1 could be more orderly. Especially add the letters in the pictures, as reported in the caption, to facilitate the reader. 
    The source is referred to as "personal file": has the patient’s consent been requested for the publication of the image? I suggest eliminating the details of the photo that could allow the recognition of the person, like the eyes.
  3. Introduction: what is meant by "condition of the target?" perhaps the optical properties of a specific biological tissue? If so, I suggest speaking of "optical properties". Otherwise indicate some of the conditions.
  4. Recheck the text. In particular, always write in the same way "blue light" (sometimes it is uppercase) and "comedones".
  5. In the clinical studies you have reported, is it possible to write who or how skin improvements have been evaluated following the blue light treatments? in my opinion is an important information that adds value to the result obtained in the study.
  6. 3.1correct the title
  7. I suggest that authors report the bibliographic reference every time an author is quoted

Author Response

Dear reviewer,

Thank you very much for your comments and I am available for questions.

Sorry for the delay, because is a holiday in Brazil and the translator delivering the manuscript review.

Firstly, we would like to thank the reviewer for all the considerations regarding our article.

Question 1 : In table of dosimetry, there is no records of the spot size diameter of the laser and the number of the cluster in the LED device, whether one wavelength or two wavelength. Could you please address this.

Answer:

Thank you for your remarks. Most of the articles reviewed do not mention this kind of data, as stated in the main text and in the conclusion. I have added a column about this to the table.

Question 2 : It would be useful for the reader to identify which type of ance vulgaris was better responder to blue light

Answer:

The articles do not allow define this positively, but inflammatory acne probably shows a better response to the blue light. This has been included in the manuscript and is highlighted in yellow in the comments.

Question 3 : Have the studies reported any residual scars after treatment? Please address this

Answer:

The answer has been highlighted in the text.

Question 4 : No follow-up reported in the tables for the selected studies. Please address this.

Answer:

The follow-up reported by the authors is shown in table 2, labeled as follow-up. This information has also been highlighted for better understanding.

Question 5 :  Mentioned the number of the sessions but it is not clear what is the time interval between sessions. Please address this

Answer:
This information has been highlighted in table 2, as Duration of exposure (weeks).

Thank you for your consideration.

Sincerely,

Mara Lúcia Gonçalves Diogo

Very grateful.

Mara Diogo

Round 2

Reviewer 1 Report

The authors have added the missing information in the manuscript, giving a clarity to the readers. I have noted the references have double numbering.  This needs to be addressed prior to publishing.